# Research Progress, Hotspots, and Trends of Using BIM to Reduce Building Energy Consumption: Visual Analysis Based on WOS Database

**DOI:** 10.3390/ijerph20043083

**Published:** 2023-02-09

**Authors:** Xun Liu, Zhenhan Ding, Xiaobo Li, Zhiyuan Xue

**Affiliations:** School of Civil Engineering, Suzhou University of Science and Technology, No. 1 Kerui Rd., New District, Suzhou 215000, China

**Keywords:** BIM (building information modeling), building energy consumption, scientometrics, Citespace, review

## Abstract

With the development of engineering technology, building information modeling (BIM) has attracted more and more attention and has been studied by many experts on building energy consumption in recent years. It is necessary to analyze and forecast the application trend and prospect of BIM technology in building energy consumption. Based on 377 articles published in the WOS database, this study adopts the technique of combining scientometrics and bibliometrics to obtain relevant research hotspots and quantitative analysis results. The findings demonstrate that the building energy consumption field has made extensive use of BIM technology. However, there are still some limitations that can be improved, and the use of BIM technology in construction renovation projects should be emphasized. This study can help readers better understand the application status of BIM technology and its trajectory of development with regard to building energy consumption, providing a valuable reference for future research.

## 1. Introduction

Emissions of greenhouse gases caused by human activity are the main reason for global warming. It is known that the construction industry accounts for 40% of world energy consumption and CO_2_ emissions, and research on reducing building energy consumption and greenhouse gas emissions has grown over the years. Meanwhile, with the comprehensive development of informatization in the 21st century, BIM as a revolutionary concept has been widely addressed. The term BIM attracted the interest of Autodesk, who started to market BIM technology and related software products. The concept of BIM was presented as a technology for the construction sector that could improve efficiency and reduce costs during the various stages of construction management [1]. After nearly ten years of rapid development, BIM technology has shown strong capabilities in various building fields, including building energy consumption.

The majority of studies can be grouped into the following categories: The first category is the analysis of energy consumption during the construction cycle. Energy consumption research is refined to each stage from the perspective of the whole life cycle of the building. The second category is studying the energy consumption reduction by analyzing the types of the buildings and their structural orientation. The third category is about user behavior analysis that examines how building users affect energy consumption. Through influencing factors analysis, methods to reduce energy consumption and their applications in engineering practice can be addressed. The fourth research category connects BIM and traditional energy consumption management software systems or other technologies. Overall, BIM technology can be used to develop third-party plug-ins or build energy consumption management platforms for practical applications.

Firstly, in the analysis of how building cycle affects energy use, Xu et al. found the causes of the building energy consumption performance gap (BEPG) and developed a BIM framework for the whole life cycle to systematically improve building energy efficiency [2]. Gan et al. proposed the method of applying computer technology to the construction industry to decrease energy use throughout the life cycle of structures [3]. Belussi et al. proposed the introduction of renewable resources into buildings, resulting in the concept of zero-energy building (ZEB) [4]. On this basis, EI Sayary and Omar applied BIM technology to zero-energy buildings. They believed that BIM should not only be used as software for assessing the consumption of energy in buildings and established a template for calculating energy consumption through BIM ideas to control energy consumption in the early design process of buildings [5]. Wei and Chen found that combining value engineering and BIM techniques helps improve design optimization and reduce building energy consumption at the design stage [6]. Zhang, C et al. integrated hidden energy consumption in the process of building construction and transportation into a BIM platform and connected the BIM model with an external database. Through analyzing different combinations of various resources, a lower energy consumption can be achieved [7].

Secondly, in terms of architecture itself, one important research aspect is improving building envelopes to improve energy efficiency [8,9,10,11]. David Bienvenido-Huertas found that the thermal transmittance of the walls is critical in the energy consumption of the building envelope [12]. Guo and Wei discovered increasing building energy savings by reducing the design cost of the building envelope [13]. Jeon et al. used BIM technology to simulate the energy of building maintenance structural units and determine how building conditions affect energy use [14]. For research on energy consumption reduction in transforming building envelope structures, many scholars have considered the integration of BIM technology [15,16,17,18]. Abanda and Byers investigated how building orientation affects building energy demand by using Revit, a BIM technology tool, and Green Building Studio, an energy simulation software [19]. Javier et al. employed BIM technology to enhance hospital buildings’ energy management systems [20]. Stegnar and Cerovsek applied BIM technology to an office building project, effectively simplifying the energy-saving transformation process of the office building [21].

Thirdly, in terms of user behavior analysis, by modifying residents’ behavior at the operational stage and applying BIM technology to facility management at the operational stage, Franeisco et al. sought a new direction in reducing building energy consumption [22]. Another area of interest by Mataloto et al. was the effect of user activity on energy consumption in buildings. They suggested using BIM data to help users understand environmental conditions in order to change user behavior and decrease energy use [23].

Finally, in terms of the combination of BIM technology and new technology, Chong et al. combined BIM with a building energy management system (BEMS) for Bayesian calibration and found that BIM technology was significantly helpful for the BEMS system [24]. Kim et al. applied BIM technology to building energy analysis (BEA), which successfully increased the possibility of exploring different solutions in the BEA process [25]. Verdaguer et al. conducted a life cycle assessment (LCA) based on BIM technology to assess environmental and energy consumption impacts [26].

It is difficult to comprehensively summarize the development and application trend of BIM technology in the improvement of building energy consumption. Many scholars applied the method of scientometrics to sort the knowledge of the whole subject field [27,28,29,30]. Fetrati Mahdieh A and Hansen Davidand Akhavan Payman used scientometrics in the field of organizational creativity and addressed the method of managing organizational creativity [31]. At the same time, there are a large number of studies using scientometrics to study the application of BIM technology in various fields [32,33]. Scientometrics is also used to analyze the research field of BIM technology separately [34,35,36]. Therefore, it is crucial to thoroughly examine how BIM technology is used in the sector of building energy consumption. Citespace and Vosviewer are used to perform bibliometric analysis and screening of the Wos database, based on which a literature review analysis was performed to identify the most active research areas and research methods. In addition, some possible research objects, research methods, research directions, and research challenges are determined based on the results of the scientometrics analysis.

## 2. Materials and Methods

In recent years, scientometrics has gradually become one of the common methods used by researchers and universities to evaluate scientific research performance. This work adopts the method of bibliometric analysis and scientometric analysis. Firstly, the appropriate database and related literature are selected through bibliometric analysis, and then the scientific econometric analysis is carried out. Before the scientometric analysis, the significance of bibliometric analysis is to evaluate the relevant journals and find the appropriate literature, so as to make the results of scientometric analysis more credible. The present study will use scientometrics methods to analyze the application of BIM technology in the field of building energy consumption. Based on series analysis, the research hotspots, the most updated research, and historical changes of the use of BIM in the field of building energy consumption will be analyzed. The research process adopted in this paper is shown in Figure 1.

### 2.1. Research Methods

Scientometrics is an emerging subject of quantitative research. Its research content describes the process of scientific development, reveals the internal mechanisms of scientific development, predicts the trend of scientific development, and provides a support basis for scientific management. The research method is mainly quantitative analysis. For research and analysis, Citespace and Vosviewer software are used in this study.

CiteSpace software was developed by Dr. Chaomei Chen, a Chinese scholar at the School of Information Science and Technology at Drexel University. It is used to assess the prospective knowledge in scientific analysis. By using visualization, it may demonstrate the internal logic, structure, and distribution of scientific information. In this study, Citespace was used to export a picture network of keyword cluster analysis and journal publishing areas and so on, so that the quantitative results based on data analysis could be used for subsequent qualitative research. Vosviewer is a document analysis and visualization software developed by the Technical Research Center of Leiden University in the Netherlands. The main advantage of Vosviewer over other document metering software is its powerful graphical display capability. This makes it suitable for processing large amounts of data. This paper uses Vosviewer to derive visual images of quantitative analysis including keyword co-occurrence analysis and author co-citation networks. By combining Citespace with Vosviewer, BIM technology in building energy consumption is comprehensively discussed and analyzed.

### 2.2. Data Sources

The current favorite source of cited data by many researchers is the Web of Science (WOS) database, which covers more than 10,000 topics and more than 10,000 subfields. WOS covers the majority of pertinent papers written by top academics and has high impact and a broad international scope. In order to ensure the reliability and comprehensiveness of data sources, this study uses the WOS database as a data source.

In this paper, building energy consumption and BIM were selected by the keyword retrieval method, 491 search results were obtained from the WOS database. Keyword search results were not filtered, and journals, conferences, and books were all included to allow for a more comprehensive study of the subject area. There was no limit on the collection period, but the collected articles are from 2010 to 2022, demonstrating that the first publication on how BIM technology is being used to reduce building energy usage began in 2010. To further guarantee the validity of data sources, the language was set as English, the WOS core collection was screened out, and the repetition was screened out through Citespace. Finally, 377 WOS retrieval results were obtained.

## 3. Results

This study carried out publication analysis, author co-citation network, and keyword analysis. Through these analyses, the current state of the application of BIM technology in the areas of building energy consumption, defective parts, and future trends is explained.

### 3.1. Analysis of Published Literature

#### 3.1.1. Analysis of the Number of Journal Publications

The selected literature has been published in a variety of internationally renowned journals, demonstrating how BIM technology is being used in the field of building energy management and has now become a wide and interesting topic in many journals. Figure 2 and Table 1 summarizes the number and percentage of journal publications according to the bibliometric records of the WOS core collection database. According to the retrieval results of the WOS, *Sustainability* has published 31 articles related to this field, ranking first. *Automation in Construction*, *Energy and Buildings*, *Journal of Production*, and *Energies* are all leading journals in the field.

#### 3.1.2. Analysis of an Annual Number of Documents Issued

After WOS database collection and post-processing, the annual number of papers on BIM technology and building energy consumption was collected. Figure 3 describes the research on BIM technology application in building energy consumption from 2011 to now. It can be seen that starting from 2011, the number of publications continuously increases, which shows that the academic community is becoming more and more interested in using BIM technology to reduce construction energy usage.

As shown in Table 2 and Figure 4, between 1994 and 2000, several publications produced studies on the subject of building energy consumption. Between 2000 and 2010, there were half the number of publications as there were in the prior ten years. However, 41% of the reviewed articles were published in 2010 or later.

As shown in Table 3 and Figure 5, the number of publications related to the topic of BIM has clearly increased since 2012 as a result of the expanding use of BIM technology in engineering practice and the ongoing promotion of various software solutions. Every year from 2016 to 2019 saw the publication of more than 600 journal articles, and this trend was maintained until 2019. It indicates that the research on BIM has entered a stage of rapid growth [36].

The research on the two fields shows that it is an inevitable trend to combine BIM with building energy consumption study.

#### 3.1.3. Regional Analysis of Periodical Publishing

In this study, CiteSpace was used to establish a research image of BIM technology in the field of building energy consumption in different countries, so as to explore the spatial distribution of relevant articles. The network has 18 nodes and a sizable number of linkages, as depicted in Figure 6, the size of the node indicates the total number of articles published in the country in 2011 and 2022.

At the same time, the number of relevant studies published in different countries is derived from the WOS database and listed in Table 4. The table shows that the top countries are China, the United States, Australia, Spain, South Korea, and Italy, demonstrating that these nations have contributed significantly to the field’s research. The United States was the first to carry out relevant research in 2011 and took second place in the number of papers that have been published, while China started relatively late and started relevant research in 2014. However, the number of published papers in recent years is so large that it has reached the first place. The number of relevant articles in these countries/regions is considerable, indicating that these countries/regions have carried out in-depth studies on the use of BIM technology in building energy.

As shown in Figure 6, China, the United States, Australia, and several other nations have a high degree of intermediary centrality (indicated by the circle size), which means that research institutions in these countries collaborate very frequently. The bigger the ring, the more academic papers were published in the country, and the darker the color, the earlier the paper was published. Table 4 makes it abundantly evident that when compared to other countries, China started studying the use of BIM technology to reduce building energy consumption recently. However, in the past few years, China has developed rapidly and become the country with the most relevant papers published.

#### 3.1.4. Most Cited Publication

Through selecting “cited literature” by Citespace, the studies that are highly cited in the research field can be found. To a certain extent, it can reflect the development context and knowledge research basis of BIM technology in the field of building energy consumption. Table 5 shows the identified highly cited classic literature with deep influence on the area of study.

Through an in-depth exploration of these five classic pieces of literature, the outstanding contributions of experts and scholars in the BIM field are analyzed. It is concluded that Wong and Zhou [37] found that most studies on BIM technology for green buildings focus on the functional improvement of the surrounding environment in the design and construction stages. To achieve more efficient low-carbon management, the integration of BIM techniques into the facility operation and maintenance in future green systema is suggested. Soust-Verdaguer et al. [26] revealed that implementable technologies are needed to reduce buildings’ energy consumption and environmental issues. They combined BIM with LCA, optimized output data, and produced better outcomes when applying BIM to building LCA. Chong et al. [38] found few studies that thoroughly reviewed BIM standards or guidelines and their sustainability uses. Then, the most advanced sustainable development of BIM is reviewed comprehensively. The results show that there is little work on the application of BIM technology in renovation and demolition, while some important insights and implications are found. Abanda and Byers [19] revealed that BIM technology is rarely considered in recent studies on the effect of building orientation on energy demand. Therefore, the research is divided into three steps: modeling, simulation, and application. Studies have shown that a good orientation can save significant amounts of energy throughout the life cycle of a building. Pan and Zhang [39] systematically reviewed the application status of artificial intelligence in the context of construction engineering and management (CEM) from both scientometrics and qualitative analysis, and discussed its future research trends.

### 3.2. Author Co-Citation Network

Considering a large number of co-cited authors, this paper selected the top authors to screen out important authors in the research field and their co-cited relationships to guarantee the accuracy of the research findings. After screening, the 45-node co-citation network of the author was discovered, as shown in Figure 7. The number of papers that authors have published is indicated by the size of the nodes. Nodes in the network are evenly distributed, and a small number of nodes are large, which indicates that the research base in this field is relatively concentrated.

Several closely linked partitions in Figure 7 are colored to indicate that there has been a strong collaborative relationship among the authors in these partitions, such as Haddad Assed, Mohammed K and Ahmed W.A, Lai Xulu, Li Zhengdao, and Xu Xiao Xiao. The central authors of these partitions can also be determined by analysis. The central authors of the study group are more cooperative than other researchers. For example, Thomas is the central author of a research group that includes Li Junjie, Yang Yifan, and Li Hongyang. In addition, the first four central authors are all from developed countries, although more authors from different regions can be found as the list of authors continues [40]. The diversity of the authors’ locations can prove that the application of BIM technology in the field of building energy consumption has received worldwide attention.

### 3.3. Keywords Co-Occurrence Analysis

Keywords are the essence description of the research article. Research hotspots in the subject of understanding building energy use can be examined by keyword co-occurrence analysis [28]. By studying the relationship between these hotspots, the past development process and future development direction in the area of building energy consumption can be displayed. By examining the BIM technology’s trend of use in the area of building energy consumption, it can be concluded that relevant papers began to surge after 2016. Therefore, for the 377 papers screened by WOS, this paper uses Vosviewer scientific metrology software to conduct keyword co-occurrence analysis on the articles after 2016. The outcomes are displayed in Figure 8. To be able to ensure the reliability of the literature analysis, this paper selects the keywords that have been cited at least ten times. All the 69 keywords that have been cited more than 10 times are added to the keyword co-occurrence analysis. Each node represents a keyword that has been cited more than ten times, and the node’s size indicates how frequently the keyword appears in the article. At the same time, the links between keywords indicate the frequency of occurrence in article co-occurrence. This network can represent the front-and-center trend of BIM technology in the research field of building energy consumption.

According to Figure 8, the top five keywords in frequency are architectural design, energy utilization, energy efficiency, sustainable development, and information theory. The published literature on BIM technology in the area of energy consumption analysis uses these keywords most frequently, and they are also related to one another.

(1)Architectural design: Architectural design is a complex task in which the design team tries to balance opposing parameters that are subject to various constraints [41]. In the area of building energy usage, this is still true. Despite advancements in building technologies and materials, building design decisions are the key to successful building energy reduction [42]. The introduction of the use of BIM technology in the stage of architectural design has been widespread.(2)Energy utilization: When the construction sector develops in the future, the active development and utilization of new energy to achieve energy-saving designs is the only way forward [43]. The direction and focus of research in the area of building energy consumption have always been on how to utilize fewer natural resources and more clean energy to reduce environmental harm.(3)Energy efficiency: It may be claimed that improving energy efficiency is the key to reducing building energy consumption because it is a significant contributor to society’s terminal energy consumption [44]. The construction industry and academia have become more and more interested in how to increase energy efficiency. The application of BIM technology into building energy management systems and maximization of the benefits of BIM technology to improve building energy conservation is a new research hotspot.(4)Sustainable development: Sustainable development is also a commonly used keyword in the field of building energy consumption, building energy conservation is definitely part of sustainable development [45].(5)Information theory: The use of information technology and its advancement in the discipline of engineering management have accelerated the modernization process of the field of engineering management and the whole construction industry. In the field of building energy consumption, information theory is often mentioned.

In addition, the keyword frequency of all 377 articles retrieved through CiteSpace is shown in Table 6. From Table 6, it can be concluded that “BIM” appears most frequently as a document retrieval keyword, reaching 42 times, and “performance”, “consumption”, and “design” rank second with 38, 38, and 37 times, respectively. Combining Table 6 and Figure 8, it is not difficult to find that most of the relevant studies are still in the design stage.

Figure 9 shows the top ten most powerful highlighted keywords, with red on the graph indicating that the keywords were highlighted during this period. The keywords are from 2011 to 2022, and in a short amount of time they have attracted much interest from the architectural science community. Words such as “BIM”, “thermal comfort”, and “sustainable design” are the most popular research hotspots in this research field. At the same time, thermal comfort and natural ventilation began to explode in 2021. There is no indication of whether the four words’ explosion phase will end. Since the data only cover research up to the end of March 2022, it is obvious that these two words’ explosion cycle will continue. As a result, it is possible to identify these keywords as the popular ones in the field of building energy consumption research using BIM technology.

These keywords have recently been highlighted in the literature examined for the purpose of this study, and they are still generating a lot of research interest. The word “thermal comfort” and its similar forms is often observed in bibliometric records. Many studies, such as Gan et al. [46], Natephra et al. [47], Zahid et al. [48], and. Shahinmoghadam et al. [49] explored how to use BIM technology for heat treatment in buildings, and evaluated or compared the impact of different buildings on energy supply and demand caused by building structures with different thermal performance. Using BIM technology to analyze heat energy consumption is among the most popular study goals of specialists in this area. In such articles, the term “IOT (Internet of things)” is frequently used. The IOT refers to the process of providing an interconnected network environment through various information carriers, such as the Internet, for the objects that can be connected, so as to obtain the interaction between these objects in real time.

When there is no time span to check the co-occurrence network of keywords (Figure 9), there is a certain coincidence with the keywords highlighted in the most commonly used keyword list. Therefore, it can be explained that some of the most commonly used keywords dominate the research field throughout the time frame of this study (2011–2022).

## 4. Discussion

By using “Burst Terms” from the CiteSpace research literature, the keywords that used to be studied, the developing ones, and those emerging ones are obtained, as shown in Figure 10. The higher the strength of the keyword, the more attention it will receive from researchers within the time interval marked red in the figure, reflecting the research frontier and hot spot of BIM technology in the field of building energy consumption from 2011 to 2022. The timeline view shows how keywords have evolved through time and how they have coexisted with different words. At the same time, “Burst articles” can also describe the hot areas in the field of research throughout a specific time period. Therefore, this paper combines the “burst terms” with the “burst articles” to better summarize some important issues in the application of BIM technology in building energy consumption. In this study, keywords are clustered by time and divided into four categories. Further discussion is as follows.

A (BIM): From 2012 to 2015, the research in this field was just emerging, and most articles were discussing the possibility of combining BIM technology with building energy consumption. For example, Wong, JKW [50] shows that through the experience of the LEED (Leading in Environmental Energy and Design) project in the United States, there is great potential in combining BIM technology with building assessment. They explored the possible use of BIM in a sustainable-building-certified residential building project in Hong Kong through a Delphi study and case study. The building model fully incorporated and employed by BIM technology is discovered to be highly complex [51]. High levels of energy performance can only be attained by constructing multi-objective design optimization on the complicated building model. This study proposed a BIM-based performance optimization integration framework—BPOpt—and gave an application example. Gokce and Gokce [52] found a lack of effective technologies for managing and monitoring buildings, and in the study, the building’s management has been severely constrained. The study addresses the issues of creating comprehensive information control through extensive application of BIM tools and non-information conventional instruments. The system’s dependability was subsequently confirmed in a research facility.

The above documents from 2012 to 2015 have built a framework or system through BIM technology to carry out case applications in the field of building energy consumption, which is still a mainstream research focus in recent years [53,54,55]. It is also noted that there is not an application framework or system for the use of BIM technology in the area of building energy [56,57]. Therefore, in future research, studies can focus on how to develop a standardized BIM system to solve most engineering energy consumption problems as the main direction. This paper puts forward several views.

(1)The construction of a BIM standardization system cannot be separated from the support of the government. At present, the government often pays more attention to policy and personnel factors but ignores the influence of economic factors. Therefore, relevant government departments should consider the input cost and economic benefit of BIM technology application in building energy consumption enterprises when formulating policies. At the same time, some incentive and tax preferential policies can be formulated to reward and promote key and difficult projects.(2)When formulating policies, the government should adopt differentiated policies for enterprises of different natures and sizes, for example, increasing support and incentive policies for small- and medium-sized enterprises that employ BIM technology to reduce emissions and conserve energy, and selecting some large enterprises that use BIM to reduce building energy consumption to promote and set examples.(3)Enterprises’ cognition on BIM technology is also very important. To apply the BIM technology in construction energy filed, improvements in the cognition of employees and senior management personnel is necessary. It is significantly important that senior managers and employees can understand the future value of applying BIM technology to reduce building energy consumption.

Therefore, relevant associations, institutions, and government departments must strengthen the education, training, and publicity of BIM technology in the field of enterprise building energy consumption. The key to the implementation of BIM technology is to improve employees’ understanding on BIM technology and obtain relevant supports from senior management departments, especially enterprise leaders. The application of the new information technology in green buildings can attract more young talent, and the increase in such corporate labels and corporate culture is also of great benefit to the enterprise itself.

B (energy simulation): From 2016 to 2017, the research upsurge in this field gradually transformed into energy simulation, that is, the model established by BIM technology is imported into energy simulation software, such as Green Building Studio and Modelica, and transformed into a BEM (building energy model). The parameters of the transformation model were analyzed to observe and detect the building energy consumption. Take the study by Abanda as an illustration, by applying Revit modeling and importing it into energy simulation software Green Building Studio, the research demonstrated that a building’s orientation has a substantial impact on the energy consumption [19]. Jeong et al. [58] proposed a framework that combines BIM and BEM based on object-oriented physical modeling, and the automatic conversion of the BIM model into a building energy model based on Modelica and thermal simulation was also investigated. The simulation results can be exported through visual components for architectural designers to view at any time. By viewing thermal simulation, architects can adjust the overall building energy consumption problem. Choi et al. [59] aims to increase the interoperability of BIM-based EPAs by creating a system with assistance from EPAs (energy performance assessments). A case study simulation experiment is run concurrently to test the system’s reliability.

There are still a lot of similar case studies using software modeling; however, the following need to be considered: firstly, how to carry on the case study research without the corresponding software; secondly, the demand for the parameter set up is strict for the model software; thirdly, the modeling format (IFC) of Revit software is separate, but its compatibility with various software is quite different. These reasons will lead to the application difficulties of BIM technology to large-scale building energy consumption case studies. This essay examines the following issues in order to address the aforementioned issues:

(1)At present, there is still no suitable software that can widely apply the IFC format of BIM modeling to the field of building energy consumption. Various commonly used software support the IFC format at different degrees, which brings challenges to researchers. This puts forward higher requirements for software companies in the area of energy use in buildings. There is an urgent demand in the next few years to produce building energy consumption simulation software that can be widely used in the IFC format.(2)Because green modeling software is not yet mature, the early stage of the BIM model has a high set of requirements, but this leads to three issues. First of all, the parameters need to be set for BIM models in the later stage of project construction. The second concern is that for designers who seldom understand green buildings, the preliminary modeling requires a lot of time and cost for them to learn, which will also affect the entire working efficiency of enterprises. So, enterprises need to integrate the ideas of green building project design and management personnel training for energy conservation and emissions reduction early in the design phase, so as to greatly reduce repeated work and decrease the cost of energy consumption.(3)The technical problem is that there might be inaccuracy of automatic translation on normative articles by computer. Even with the aid of artificial intelligence, there is still a problem that the training model is not perfect due to the lack of data sets. Therefore, the translation work is still performed manually. Subsequent research work can achieve automatic translation by improving the accuracy of computer translation.

C (multi-object optimization): Since 2018, articles on the use of BIM technology in the field of building energy consumption expanded. Researchers considered the integration of multiple factors into the BIM information model for objective optimization. Shadram and Mukkavaara [60] used a multi-objective optimization model to study the balance between operational and consumption energy and integrate this operation mode into the BIM information model to further strengthen operability through BIM visualization technology. Their research proposed a research framework combining a multi-objective optimization model and BIM information technology and verified the reliability of the framework. Yang and Liu [61] developed an optimization system based on BIM and BPS technology. NSGA-II genetic algorithm was used to achieve multi-objective optimization of CO_2_ emission and discomfort index, which could effectively improve building lighting conditions and reduce residential CO_2_ emission. At the same time, the case is applied to the early decision-making stage of housing energy saving and emission reduction. Zhang et al. [7] pay attention to how much energy is used during material transportation. Although the environmental impact of construction material transportation is small compared with that of the operation stage, if it is allowed to accumulate continuously at the national level the impact will be very large. In order to reduce energy consumption throughout the building process and incorporate implied building energy usage out of the BIM platform, adjustment of the quantity and capacity of involved setups is needed.

So far, there are many case applications in the published studies which involve the solutions and methods of using BIM technology for energy consumption management in various processes within the whole life cycle of construction. However, most studies still focus on using BIM technology to reduce building energy consumption in the process of building design [62]. In reality, traditional building energy management is involved after the structure is constructed or during the operational phase [63]. However, BIM technology has certain virtual decision-making abilities in the later operation stage, but it still has great limitations. This is reflected in the early design that designers should pay attention to parameter setting. In order to facilitate its use in the later operation stage, the model information in the early design stage should be ready to be transferred to the later operation stage. Transportation of data carriers deserves further study by scholars.

D (building renovation): Since 2018, researchers found that building energy consumption increased year after year, making retrofitting existing buildings an urgent need. In 2019 and 2020, many researchers provided more suggestions for reducing energy consumption during remodeling old buildings, Freitas et al. [64] studied how to integrate solar power design tools in the BIM environment in the initial stages of a building’s design. Through the case study, building integrated photovoltaic (BIPV) systems were addressed to transform the facade and roof of existing office buildings in order to lower energy consumption. Gan et al. [46] proposed a framework combining BIM and machine learning technology to detect the optimal thermal condition of indoor environments. Through case analysis results, it is proved that rational use of natural ventilation can reduce the amount of energy used for thermal comfort in buildings, but it cannot meet all thermal comfort standards. However, different energy consumption levels required for refrigeration varies by season. The study found that the energy used for refrigeration can be effectively reduced by measuring the system’s thermal comfort conditions at different seasons, and this system can be applied to most existing building renovations with refrigeration requirements. Joblot et al. [65] reviewed that in recent years, companies engaged in renovation work did not use BIM tools in building renovation work, which would not only cause a certain amount of energy waste but also affect the efficiency of renovation project reuse.

Through a large literature review, it is found that in recent years, many scholars are conducting research on building renovation. However, most of the studies focused on the impact of large-scale renovation plans on the national level, rather than taking into consideration new technologies when renovating buildings. Therefore, this paper puts forward the following suggestions:

(1)Researchers should pay attention to the use of BIM technology in building renovation projects and consider the challenges and stumbling blocks of using BIM technology in building renovation. At present, research on using BIM technology in the reconstruction of the building envelope is rising. For example, for the improvement of building materials, researchers should also try to use BIM technology for the management of building renovation. Rational use of BIM technology can greatly reduce the energy consumption caused by improper management in the construction and operation stage.(2)Enterprises related to building renovation should focus on the implementation of green environmental protection and information technology and increase public knowledge of building energy use by training employees and communicating with universities or scientific research institutions.(3)Governments should consider their economical and social conditions to determine the regulatory obstacles that affect the combination of building renovation technology and information technology and put forward relevant policies to solve these problems as much as possible.

## 5. Conclusions

Due to the BIM technology’s fast development, the research and engineering practices in the filed of building energy consumption has undergone significant changes in the past decade. This study combined scientometrics and bibliometrics to explore and compare the historical development and frontier direction of the application of BIM in the field of building energy consumption. A total of 238 studies from the literature on BIM technology applications in the building energy consumption field were collected from the WOS core database. Through keyword cluster analysis and keyword co-occurrence analysis, the research history and current situation of this research field were determined. Then, combined with author co-citation network analysis and journal publishing field analysis, the global research hotspots in this field and the research teams with the most research in this direction were identified, and the number of journal publications was studied as well. Through the above scientific econometric analysis, the following conclusions are drawn:

(1)First of all, research direction and hotspots of BIM technology in the field of quasi-energy consumption were determined through network of co-concede keywords analysis and keyword cluster analysis. The evolution of keywords in this field was studied based on the time factor. According to the research, the common keywords in 2015 were building information modeling and sustainable design. Later, the keywords evolved into thermal comfort and natural, these keywords are also the research hotspot for the next few years.(2)The most influential institutional authors and their partnerships are also discussed. Hadded, Li Zhengdao, Vivian, Martin, and S. Thomas are found to be major contributors to this research field, and the analysis of the research team composed of researchers in this field shows that the major contributors to this research field are not closely connected.(3)In terms of publishing frequency, *Sustainability*, *Automation in Construction*, and *Journal of Cleaner Production* are the top three journals with the most research papers published. In terms of influence, *Automation in Construction*, *Energy and Buildings*, and *Journal of Cleaner Production* are the leading three. The publication frequency of *Sustainability* is not proportional to its influence. This means that even if articles are published more frequently, it does not necessarily mean that those journals are more influential.(4)The analysis of the publication year of the literature shows that, in terms of publication volume, the use of BIM technology in relation to building energy use has gone through three stages: conception, diffusion, and explosion. In 2011, the number of papers published was low, indicating that the field was still in the conceptual stage, but it has seen an explosion in publication since 2019. It can be seen that BIM technology has a very broad research prospect in the area of energy use in buildings. In terms of the distribution of countries and regions studied in this field, China and the United States ranked first and second, respectively. Spain, South Korea, and Australia also have relatively stable research teams. Compared with the United States, research in this field in China started late, but it has shown explosive growth in the past five years.

Overall, the value of this study is that researchers and practitioners can easily identify the journals and research teams in this research field. Due to the limitation of the scope of the WOS, the research findings might not fully encompass all of the BIM-technology-related literature on building energy consumption that is currently available. Through the progress of scientometrics technology in the future, these defects can be remedied to some extent.

## Figures and Tables

**Figure 1 ijerph-20-03083-f001:**
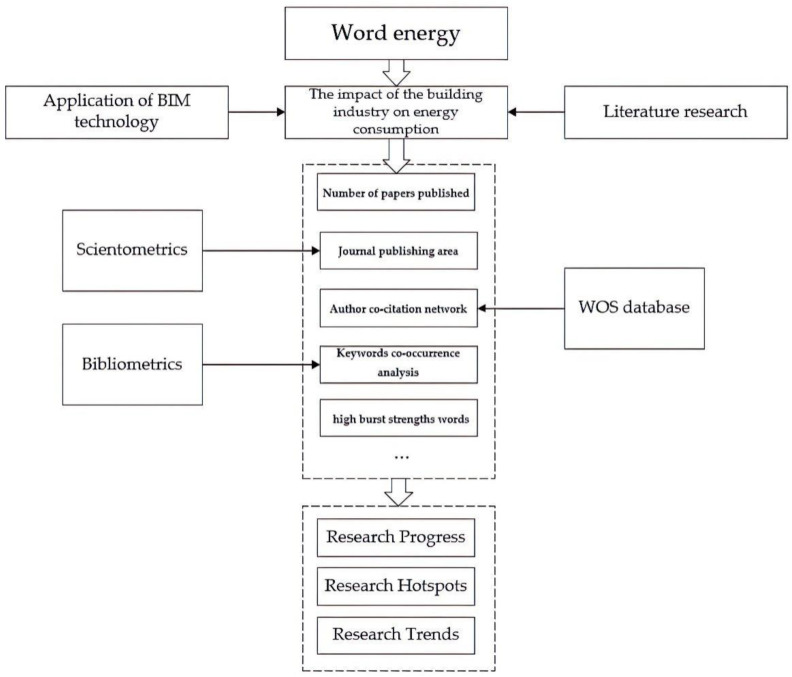
Research methodology.

**Figure 2 ijerph-20-03083-f002:**
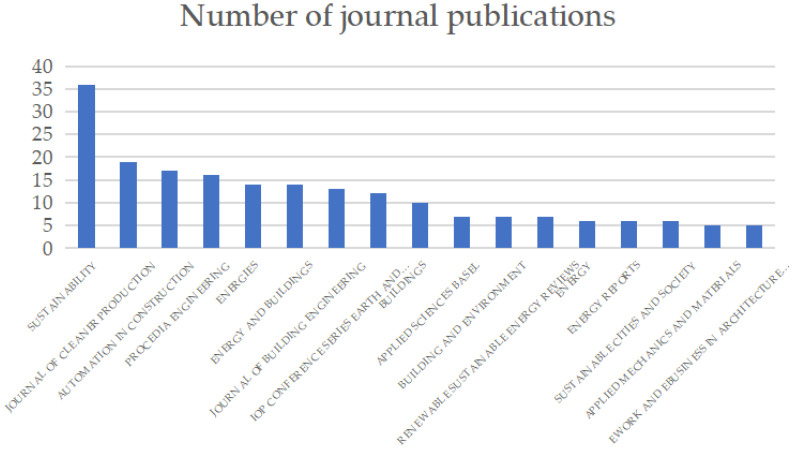
Number of journal publications.

**Figure 3 ijerph-20-03083-f003:**
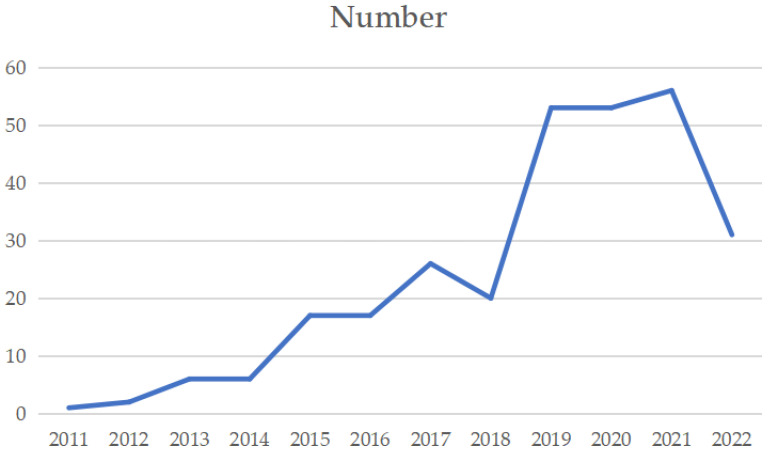
Number of papers published.

**Figure 4 ijerph-20-03083-f004:**
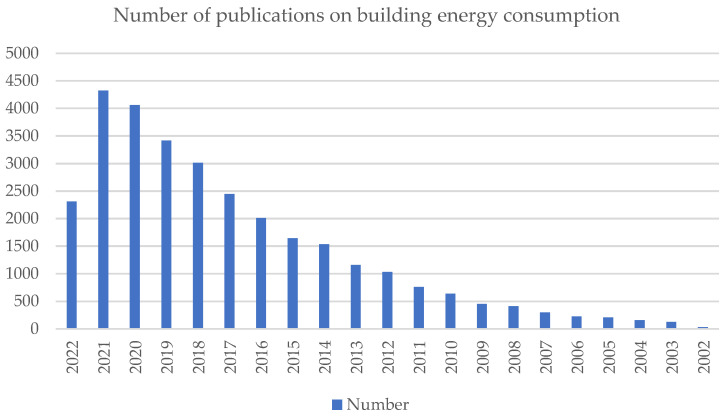
Number of publications on building energy consumption.

**Figure 5 ijerph-20-03083-f005:**
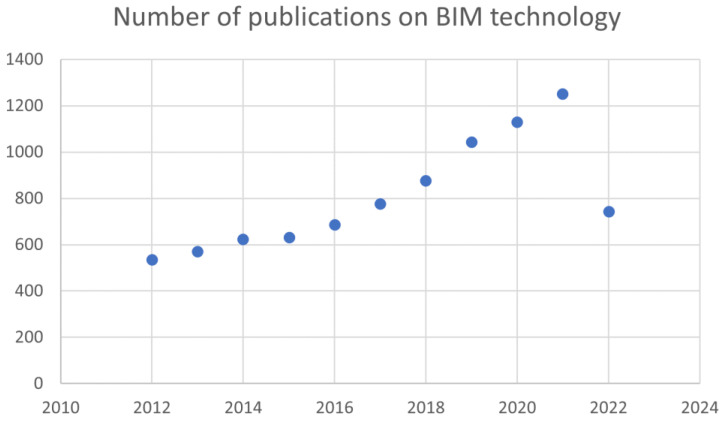
Number of publications on BIM technology.

**Figure 6 ijerph-20-03083-f006:**
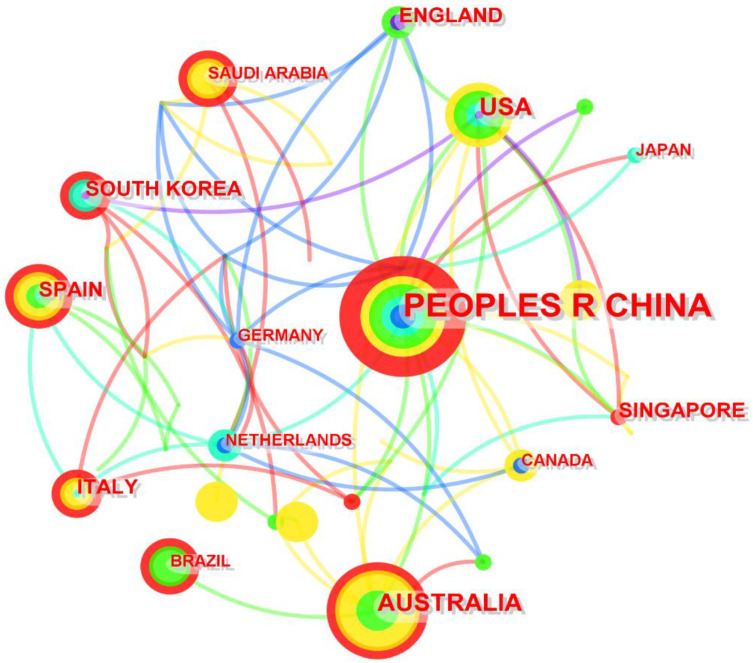
Journal publishing area.

**Figure 7 ijerph-20-03083-f007:**
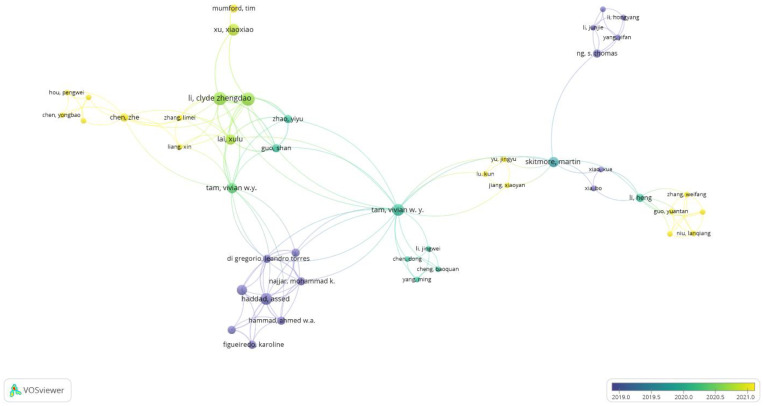
Author co-citation network.

**Figure 8 ijerph-20-03083-f008:**
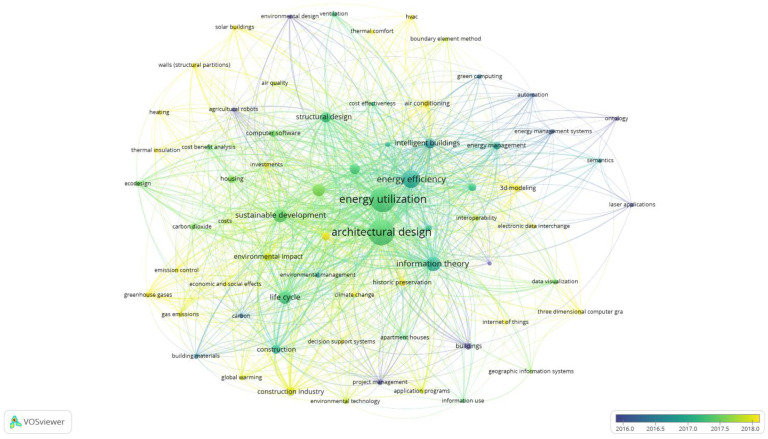
Keyword co-occurrence analysis.

**Figure 9 ijerph-20-03083-f009:**
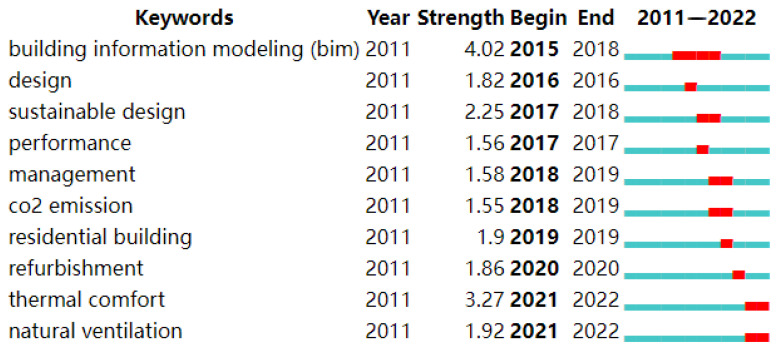
Top 10 words with high burst strengths.

**Figure 10 ijerph-20-03083-f010:**
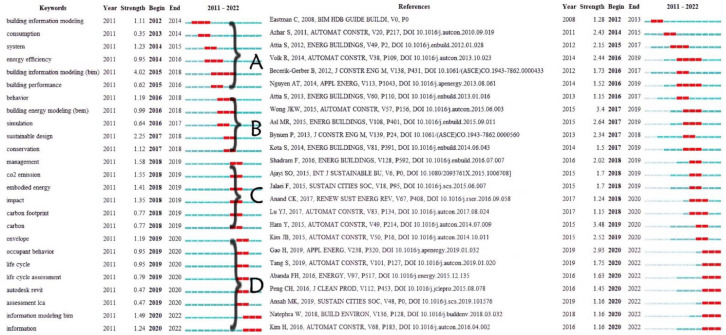
Burst terms and burst articles of BIM technology in building energy consumption research.

**Table 1 ijerph-20-03083-t001:** Number of journal publications.

Journal Title	Number of Published	Percentage
*Sustainability*	36	9.549
*Journal of cleaner production*	19	5.04
*Automation in construction*	17	4.509
*Procedia engineering*	16	4.244
*Energies*	14	3.714
*Energy and buildings*	14	3.714
*Journal of building engineering*	13	3.448
*Iop conference series earth and environmental science*	12	3.183
*Buildings*	10	2.653
*Applied sciences basel*	7	1.857
*Building and environment*	7	1.857
*Renewable sustainable energy reviews*	7	1.857
*Energy*	6	1.592
*Energy reports*	6	1.592
*Sustainable cities and society*	6	1.592
*Applied mechanics and materials*	5	1.326
*Ework and ebusiness in architecture engineering and construction*	5	1.326

**Table 2 ijerph-20-03083-t002:** Number of publications on building energy consumption.

Year	Number of Publications	Percentage
2022	2311	7.585
2021	4323	14.188
2020	4059	13.321
2019	3417	11.214
2018	3013	9.888
2017	2444	8.021
2016	2014	6.61
2015	1644	5.395
2014	1539	5.051
2013	1163	3.817
2012	1036	3.4
2011	763	2.504
2010	640	2.1
2009	455	1.493
2008	413	1.355
2007	297	0.975
2006	228	0.748
2005	207	0.679
2004	156	0.512
2003	127	0.417
2002	30	0.098

**Table 3 ijerph-20-03083-t003:** Number of publications on BIM technology.

Year	Number of Publications	Percentage
2022	743	6.166
2021	1251	10.382
2020	1128	9.361
2019	1042	8.647
2018	876	7.27
2017	776	6.44
2016	685	5.685
2015	630	5.228
2014	623	5.17
2013	570	4.73
2012	534	4.432

**Table 4 ijerph-20-03083-t004:** Number of papers published in different countries (frequency > 5).

Number	Year First Published	Countries
47	2014	Peoples R China
23	2011	USA
15	2015	Australia
14	2013	Spain
13	2012	South Korea
12	2016	Italy
10	2014	England
8	2019	Brazil
8	2018	Singapore
7	2014	Germany
7	2014	Canad
7	2020	Saudi Arabia
6	2017	Netherlands

**Table 5 ijerph-20-03083-t005:** The top 5 most cited papers.

No.	Author	Title	Resource	Cited Times
1	Wong, JKW	Enhancing environmental sustainability over building life cycles through green BIM: A review	*Automation in construction*	253
2	Soust-Verdaguer	Critical review of bim-based LCA method to buildings	*Energy and buildings*	182
3	Chong, HY	A mixed review of the adoption of Building Information Modelling (BIM) for sustainability	*Journal of cleaner production*	162
4	Abanda	An investigation of the impact of building orientation on energy consumption in a domestic building using emerging BIM (Building Information Modelling)	*Energy*	153
5	Pan, Y	Roles of artificial intelligence in construction engineering and management: A critical review and future trends	*Automation in construction*	89
6	Asl, MR	BPOpt: A framework for BIM-based performance optimization	*Energy and buildings*	89
7	Gocer, O	Completing the missing link in building design process: Enhancing post-occupancy evaluation method for effective feedback for building performance	*Energy and buildings*	87
8	Ascione F	Building envelope design: Multi-objective optimization to minimize energy consumption, global cost and thermal discomfort. Application to different Italian climatic zones	*Energy*	84
9	Wong, JKW	Implementing ‘BEAM Plus’ for BIM-based sustainability analysis	*Automation in construction*	81
10	Li, Y	Review of building energy performance certification schemes towards future improvement	*Renewable & Sustainable energy reviews*	77

**Table 6 ijerph-20-03083-t006:** High-frequency keywords (frequency > 10).

Sequence Number	Frequency	Year	High-Frequency Keywords
1	42	2017	BIM
2	38	2014	performance
3	38	2013	consumption
4	37	2015	design
5	25	2014	energy efficiency
6	25	2014	system
7	24	2013	simulation
8	23	2015	framework
9	21	2017	building
10	20	2015	energy consumption
11	19	2012	building information modeling
12	18	2018	life cycle assessment
13	16	2015	impact
14	16	2013	energy simulation
15	15	2016	construction
16	14	2019	thermal comfort
17	14	2014	building information modelling (BIM)
18	13	2016	optimization
19	13	2018	management
20	13	2015	building information modeling (BIM)
21	12	2018	energy performance
22	12	2016	model

## Data Availability

The data presented in this study are available on request from the corresponding author.

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
