# Peer review of "Research Progress, Hotspots, and Trends of Using BIM to Reduce Building Energy Consumption: Visual Analysis Based on WOS Database"

_ijerph, 2023, doi:10.3390/ijerph20043083_

Round 1
Reviewer 1 Report
Dear Authors,
The manuscript does not comply with the template. Except the email address, all other identification details for affiliation are missing, for all authors. The missing details must be corrected. It is strictly necessary to do so.
I quote from the template:
“ 1 Affiliation 1; e-mail@e-mail.com
2 Affiliation 2; e-mail@e-mail.com
* Correspondence: e-mail@e-mail.com; Tel.: (optional; include country code; if there are multiple corresponding authors, add author initials) ”
Also, the author of correspondence is marked erroneously. Same recommendation, it is strictly necessary to correct the error.
The manuscript is not flowing, it is not clear.
From the point of view of the English language, it is very confusing and the message is not understood. English must be extensively corrected overall.
Line 55: what is the role of parenthesis (1)?
For a review paper the reference list to my opinion is too short. I recommend to be enriched by adding new references.
Reviewer 2 Report
Dear Authors,
The topic for which you analyzed on scientometric and bibliometric parameters is contemporary, and as you wrote, it is expected that there will be more and more papers on similar topics from year to year. I have no complaints regarding the analysis method itself, but the clarity of the sentences must be of better quality. Some sentences are too long and unclear. Also, it happens that you start writing some parts of the text, such as a literature review, without first writing that the literature review will follow, thus creating an introduction to a part of the text. You often made spelling mistakes and started sentences with a lowercase letter. Recommendations for the correction of academic work are as follows:
Comment 1:
Line 16- “this” instead “This”.
Comment 2:
Line 29- “CO2” instead “CO2”. This is repeated in several other places in the text.
Comment 3:
Line 40- 54 – Sentences should be shorter and differently separated because this way it is much more difficult to understand.
Comment 4:
Line 50 – “Fourthly” instead “fourthly”.
Comment 5:
The text is totally unrelated. It should be noted that the literature review for each of the four criteria follows. Authors made a literature review for each of the four categories, but I think it should be defined differently, ie. to add subchapters.
Comment 6:
Line 55 - “found” instead “Found”.
Comment 7:
Line 75- “use [10].” instead “use. [10].”
Comment 8:
Line 132- “the” instead “The”.
Comment 9:
Line 181- Figure 2 is Bold. The text is in bold.
Line 193- “Table 2” and “Figure 3” . The text is in bold.
Somewhere in the text, figures and tables are written in bold letters, and somewhere they are not. Harmonize the labeling throughout the text.
Comment 10:
Match Table 3 and Figure 4 because Figure 4 also shows the year 2022, which is not shown in Table 3.
Comment 11:
Line 210- “building” instead “Building”. This is just another place where a capital letter is written instead of a small one. I wrote you some places in the comments, and you should check the entire text.
Comment 12:
Line 217- “depicted in Figure 5 The size of the node.” The period at the end of the sentence is missing or a lowercase letter is needed (between "Figure 5" and "The").
Comment 13:
Line 252- “Soust-Verdaguer” instead “Soust-verdaguer”.
Comment 14:
Line 274- The period missing - “discovered, as shown in Figure 6 The number of papers”.
Comment 15:
Line 293- “analysis.[24].” Two separate periods as a sentence break. One should have been used.
Comment 16:
Line 308- “energy Utilization, Energy efficiency, Sustainable Development, and information theory.” Should all words start with a lowercase letter?
Comment 17:
Line 316- “was introduced., and there” . Either a period or a comma.
Comment 18:
Line 317- “up to now”. The period is missing at the end of the sentence.
Comment 19:
Line 362- Figure 8 should be of better quality.
Comment 20:
Line 367- (Gan et al [30], Natephra et al [31], Zahid et al [32], and 367 Shahinmoghadam et al [33].
There is no closed parenthesis and after each author and co-author ("et al") there should be a period.
Comment 21:
Line 432 – “and sizes. such as increasing support and incentive”. Is it a full stop?
Comment 22:
Line 448 and Line 450- “BIM technology” instead “BM technology”. Is that so?
Comment 23:
Line 488- “designer, Parameter”. Is there a full stop instead of a comma?
Comment 24:
Line 537- “ 2020, Many researchers”. A lowercase letter should be used.
Reviewer 3 Report
It must be recognized that any advance in energy efficiency, in this case in buildings, is of interest. In this case, the study focuses on scientometrics and bibliometrics applied to the field of energy consumption in buildings and BIM technology, noting that there are still some limitations.
However, some improvements could be made as indicated below:
-In general, the wording and some specific issues of "English" could be improved. Also, some formal and editorial errors should be checked. For example, space between words is missing (see Keywords), or in the sentence "This article is based on 377 pieces of literature published in the Wos database. Based on 377 articles published in the Wos database, This", in which redundancies occur and there is a ", " that should be ".".
-Regarding the "Abstract" it is correct, but it could be improved by structuring it in the classic way "objects, methodology, results, discussion, and conclusions".
-Regarding the introduction, it is understood that it has ignored important authors. Some reference should be made in the introduction to the fact that great efforts are being made to improve the building envelope for energy efficiency purposes and include references to authors such as David Bienvenido-Huertas "Review of in situ methods for energy assessment thermal transmittance of the walls" and that this is also being transferred to BIM.
-References to authors who have studied scientometrics as a research methodology are also missing (Akhavan, P. among others)
- Regarding the methodology, it is understood that the wording could be improved. Even a general study flowchart would be helpful. The authors must assume that the readers must be able to reproduce as easily as possible what has been done.
-The quality of some figures such as 8 and 9 should be improved.
- Regarding the results, they are interesting, but perhaps they should be presented with more detailed limitations.
-Regarding the discussion and conclusions, ideas and problematic issues detected are exposed (standardization, exchange formats, software and its problems, etc.), but most of these are previously known issues and have been dealt with by various authors, which raises doubts about the significance of the results compared to other similar proposals.
-In short, the manuscript should be reviewed, since in addition to some formal and methodological issues, it raises doubts about whether the progress is true and significant enough, compared to other similar advances and already known issues.
Round 2
Reviewer 1 Report
The authors have made some changes but it is not enough to be in the form of publication.
Information about the affiliation of the authors continues to be missing. I don't know for sure, but maybe the authors don't understand what affiliation means, which is doubtful.
The manuscript does not comply with the template. Except the email address, all other identification details for affiliation are missing, for all authors. The missing details must be corrected. It is strictly necessary to do so.
I quote from the template:
“Affiliation 1; e-mail@e-mail.com
Affiliation 2; e-mail@e-mail.com
* Correspondence: e-mail@e-mail.com; Tel.: (optional; include country code; if there are multiple corresponding authors, add author initials) ”
The text of the manuscript has a lot of typos, therefore I suggest to the authors to check it again.
Reviewer 2 Report
Dear Authors,
Thank you for your answers! You have implemented most of my recommendations and I am satisfied with your answers. Additional recommendations are as follows:
· Still some sentences are too long which leads to the sentences being unclear, for example (Line 65-68 - Zhang, C et al. integrated the hidden energy consumption in the process of building construction and transportation into the BIM platform,the BIM model can be connected with an external database, and then lower energy consumption can be achieved by analyzing different combinations of various resources [8].)
· Throughout the text there are a lot of technical errors such as lack of spacing (for example : Line 100 – “Davidand”; Line 177- space missing; Line 220 - Figure 5.Number of publications on BIM technology ) and uppercase letters instead of lowercase letters (Line 52- Firstly, In the ; Line 70- Secondly, In terms)
Reviewer 3 Report
The manuscript has been modified by the authors and although it could be improved in some aspects, it is considered that it could be published only slightly adjusting Figure 6, since the most prominent country, its knot size, and letter name are a bit disproportionate with respect to the rest.
Sincerely
